# Role of Circular RNAs in Pulmonary Fibrosis

**DOI:** 10.3390/ijms231810493

**Published:** 2022-09-10

**Authors:** Jian Zhou, Yali Chen, Menglin He, Xuehan Li, Rurong Wang

**Affiliations:** Laboratory of Anesthesia and Critical Care Medicine, Department of Anesthesiology, West China Hospital, Sichuan University, Chengdu 610017, China

**Keywords:** circRNA, pulmonary fibrosis, silicosis

## Abstract

Pulmonary fibrosis is a chronic progressive form of interstitial lung disease, characterized by the histopathological pattern of usual interstitial pneumonia. Apart from aberrant alterations of protein-coding genes, dysregulation of non-coding RNAs, including microRNAs, long non-coding RNAs, and circular RNAs (circRNAs), is crucial to the initiation and progression of pulmonary fibrosis. CircRNAs are single-stranded RNAs that form covalently closed loops without 5′ caps and 3′ tails. Different from canonical splicing of mRNA, they are produced from the back-splicing of precursor mRNAs and have unique biological functions, as well as potential biomedical implications. They function as important gene regulators through multiple actions, including sponging microRNAs and proteins, regulating transcription, and splicing, as well as protein-coding and translation in a cap-independent manner. This review comprehensively summarizes the alteration and functional role of circRNAs in pulmonary fibrosis, with a focus on the involvement of the circRNA in the context of cell-specific pathophysiology. In addition, we discuss the diagnostic and therapeutic potential of targeting circRNA and their regulatory pathway mediators, which may facilitate the translation of recent advances from bench to bedside in the future.

## 1. Introduction

Pulmonary fibrosis (PF) is a chronic progressive form of interstitial lung disease (ILD), characterized by the histopathological pattern of usual interstitial pneumonia (UIP) [1]. Clinical manifestations of PF include increasing cough, progressively exertional dyspnea with or without dry cough, hypoxemia, and respiratory failure [2,3]. Genetic factors, aging, environmental exposures (such as smoking, silica, or radiation), and certain comorbidities (such as autoimmune/connective tissue diseases) are risk factors that may cause PF [4]. The underlying pathophysiology of PF involves dysregulated reparative response, wherein repetitive epithelial injury leads to epithelial-to-mesenchymal transition (EMT), macrophage activation (MA), and polarization, as well as persistent fibroblast activation and fibroblast-to-myofibroblast transition (FMT) and the subsequent excessive deposition of extracellular matrix (ECM) (see Figure 1) [5]. PF incurs substantial socioeconomic costs and a considerable public health burden, with a conservatively estimated prevalence of 3–9 per 100,000 individuals [6]. In addition, PF exerts significant effects on quality of life, and the median survival after diagnosis is only 3–5 years [7]. Treatment of PF is still challenging, due to limited treatment options. The only two FDA-approved drugs for idiopathic pulmonary fibrosis (IPF), pirfenidone, and nintedanib showed limited effect on decline of disease progression or mortality [5,8]. As lung transplantation is the only possible way to cure PF, it is important to identify new therapeutic targets for developing mechanism-driven treatments to tackle this refractory disease.

Circular RNAs (circRNAs) are single-stranded RNAs that form covalently closed loops without 5′ N7-methylguanosine (m7G) caps and 3′ polyadenylated tails [9]. Back in 1976, circRNAs were first discovered in pathogenic plant viroid [10]. However, they were considered to be functionless by-products of abnormal RNA splicing and did not draw much scientific attention until decades later. A timeline of significant discoveries in circRNA research is depicted in Figure 2 [10,11,12,13,14,15,16,17,18,19,20,21,22,23,24,25]. Different from canonical splicing of mRNA, they are produced from the back-splicing of precursor mRNAs (pre-mRNA) and have unique biological functions [26]. They function as important gene regulators through multiple actions, including miRNA or protein sponges or decoys, templates for translation, protein scaffolding, and recruitment or enhancer of protein function [27]. CircRNA has been reported to be implicated in many different pathophysiological processes through diverse mechanisms of action. A bulk of studies demonstrated that their aberrant expression or function are involved in the initiation and development of various diseases, such as central nervous system diseases, aging and longevity, cardiovascular diseases, kidney disease, and cancer [28,29,30,31,32]. In particular, the enrichment of circRNA was observed in the EMT [33], and accumulating evidence indicated that circRNA may play a regulatory role in PF.

It has now been realized that dysregulation of non-coding RNAs (ncRNAs) is critical to the development and maintenance of PF. Although there has been much research on mi-RNAs and lncRNAs, few researchers have taken circRNAs into consideration. With the rapid development of high-throughput sequencing technologies and bioinformatics analysis, a growing body of original articles demonstrated that circRNA expression is dysregulated in many types of PF, including silicosis (silica-induced pulmonary fibrosis), idiopathic PF (pulmonary fibrosis of unknown cause), cigarette smoke-induced PF, in which circRNA dysregulation gives rise to a complex interplay of cell types and signaling pathways, such as endothelial–mesenchymal transition (EndMT), EMT, MA and polarization, and FMT. In this state-of-art review, we comprehensively summarize the alteration and functional role of circRNAs in PF, followed by in-depth discussion of the regulatory mechanisms of circRNAs in PF in the context of cell-specific pathophysiology. In addition, we discuss the diagnostic and therapeutic potential of targeting circRNA and their regulatory pathway mediators, which may facilitate translation of recent advances from bench to bedside in the future.

## 2. Properties of circRNAs

The majority of circRNAs are formed by back-splicing of pre-mRNAs or long noncoding RNAs, in which a downstream 5′ splice donor site is joined to an upstream 3′ splice acceptor site in reverse order with a 3′,5′-phosphodiester bond at the back-splicing junction sites (BSJ) [26,34]. Back-splicing relies on the canonical spliceosomal machinery but is much less efficient than cognate linear splicing [35,36]. The looping structure that brings BSJ into close proximity is critical to back-splicing, which can be facilitated by base-paring between inverted intronic complementary sequences or dimerization of RNA-binding proteins (RBP) [9]. More rarely, if the intron lariats produced during conventional splicing avoid debranching and preserve a circular form with a 2′,5′-phosphodiester bond between the back-spliced exons, they are referred to as circular intronic RNA (ciRNA) [37] (see Figure 3).

In addition to the regulation of circRNA biogenesis, the abundance of circRNA also depends on the trafficking and turnover of circRNA. With the exception of ciRNA and exon-intron circRNAs (ElciRNA), which are limited to the nucleus, most circRNAs are predominantly located in the cytoplasm [18,38,39,40]. Nuclear export of circRNAs to the cytoplasm occurs in a length-dependent manner and requires multiple proteins, such as spliceosome RNA helicase DDX39B (for long circRNAs) and ATP-dependent RNA helicase DDX39A (for short circRNAs) [41]. After binding to transcription–export complexes, they are further recruited by the NTF2-related export protein 1-nuclear RNA export factor 1 heterodimeric export receptor and transferred into the cytoplasm through the nuclear pore complex. Moreover, circRNA shuttling also can be modulated by RNA modification or further exported to extracellular vesicles [42,43,44]. Owing to their circular structure, circRNAs are resistant to degradation by linear RNA decay machineries, making them exceptionally stable once formed (a median half-life of 18.8–23.7 h of circRNAs, compared with a median half-life of 4.0–7.4 h of their linear counterparts) [45]. So far the mechanisms of circRNAs degradation are not fully elucidated. Recent studies have shown that both miRNA and some endonucleases participate in circRNA decay [46,47,48] (see Figure 3).

Many circRNAs exert important biological functions through diverse mechanisms of action. Firstly, miRNA sponging is the most frequently proposed and extensively studied function of circRNAs. By binding and sequestration of miRNAs and preventing miRNAs from repressing their target mRNAs, circRNAs can participate in gene regulation in a post-transcriptional way [49,50]. Secondly, some circRNAs containing RBP-binding motifs can interact with the RBP and function as protein sponges or decoys, thus indirectly regulating their functions [51,52]. Thirdly, circRNA may act as scaffolds to bring enzymes and their substrates into proximity, influencing the reaction kinetics [53,54,55]. Fourthly, a few circRNA have been shown to recruit certain proteins to specific loci or subcellular compartments [56]. Fifthly, some nuclear circRNAs can also regulate gene expression by enhancing the RNA polymerase II [37,38]. Last but not least, circRNAs that contain internal ribosome entry sites can serve as templates for translation in a cap-independent manner or m6A-dependent manner with low efficiency [57,58,59] (see Figure 4).

## 3. The circRNA Expression Profiling and Integrative Analysis in Pulmonary Fibrosis

Currently, microarray or RNA sequencing (RNA-seq) analysis and reverse transcription-quantitative PCR are the most commonly used methods for the identification and validation of expression profilings of deregulated circRNAs in various disease models. Several studies have reported that many circRNAs are deregulated in various models of PF (Table 1 and Table 2; Figure 5). By Arraystar Human circRNA Microarray, 67 differentially expressed circRNAs (DECs) were identified, among which 38 and 29 were upregulated and downregulated, respectively, in the plasma of patients with IPF, in comparison with matched healthy controls [60]. Based on RNA sequencing analysis in the *murine* model of bleomycin-induced pulmonary fibrosis (BIPF), Li and colleagues found 74 DECs between PF and corresponding control tissues [61]. Similarly, a total of 10 circRNAs were detected by Hiseq 2000 from a rat model of BIPF, and there were 2 circRNAs upregulated and 8 downregulated according to the criteria *p* < 0.05 and fold change > 2.0 [62]. To elucidate the interaction networks of differentially expressed mRNAs and ncRNAs during PF process, Liu and colleagues analyzed the circRNA expression profiling from BIPF rats and control rats. The results of whole transcriptome sequencing analysis showed that 287 DECs were upregulated and 318 DECs were downregulated [63]. Aside from IPF-specific circRNAs, efforts have been made to identify circRNAs pertinent to silicosis. Cheng and colleagues performed RNA-seq screening to identify plasmatic circRNAs as potential biomarkers for diagnosing silicosis. Overall, 243 DECs were identified, of which 139 were upregulated DECs and 104 were downregulated [64]. Additionally, another study on the *murine* model of silicon dioxide (SiO_2_)-induced silicosis, using microarray analysis, identified 120 DECs (73 upregulated and 47 downregulated) between the SiO_2_ and control group [65]. Interestingly, further Kyoto Encyclopedia of Genes and Genomes (KEGG) analysis revealed that these DECs were enriched in pathways of adherens junction [60,63], purine metabolism [66], lysine degradation [63], phosphoinositide 3 kinase (PI3K)-serine/threonine Akt signaling [62], and wingless/integrated (Wnt) signaling [63].

## 4. Biological Roles and Regulatory Mechanisms of circRNAs in Pulmonary Fibrosis

### 4.1. Endothelial–Mesenchymal Transition

Apart from EMT, endothelial cells can also be the origins of the mesenchymal cells responsible for dysregulated ECM accumulation in the PF; the process through which endothelial cells transdifferentiate into motile mesenchymal cells is termed EndMT. EndMT is a cellular and molecular process through which cells lose their endothelial markers (downregulation of VE-Cadherin) and acquire the mesenchymal or myofibroblast phenotype (upregulation of α-smooth muscle actin [α-SMA] and type I collagen [Col I/COL1A1]) [67]. Fang and colleagues reported that the level of circular RNA HECT domain E3 ubiquitin protein ligase 1 (*circHECTD1*) was rapidly increased in SiO_2_-induced EndMT in both mouse endothelial cell line MML1 and human endothelial cell line HUVECs. In turn, the elevated *circHECTD1* downregulated the expression of HECT domain E3 ubiquitin protein ligase 1 (HECTD1) protein. Targeting *circHECTD1* via small interfering RNA (siRNA) or overexpression of HECTD1 by the CRISPR/Cas9 system prevented SiO_2_-induced EndMT. Moreover, consistent downward tendency of HECTD1 expression was also found in tissue samples from the mice model of SiO_2_-induced silicosis and silicosis patients, demonstrating the involvement of the *circHECTD1*/HECTD1 pathway in the pathogenesis of silicosis. Further functional experiments revealed that *circHECTD1*-mediated downregulation of HECTD1 is involved in the SiO_2_-induced EndMT by promoting endothelial cell migration and activation [82].

### 4.2. Epithelial-To-Mesenchymal Transition

During fibrogenesis, the critical myofibroblasts responsible for ECM synthesis can be derived from resident fibroblasts, circulating fibrocytes, epithelial cells, endothelial cells, and other specialized cell types. Apart from activation of tissue-resident fibroblasts and differentiation of circulating fibrocytes, EMT is an important mechanism that increases the net pool of myofibroblasts [83]. EMT is characterized by the loss of epithelial identity (apical–basal polarity and stable intercellular junctions) and acquisition of the mesenchymal phenotype (cytoskeletal and morphological rearrangements, fibroblast-like gene expression profile, migratory capacity, and ability to produce the ECM) [84]. As an embryonic process, EMT is not only involved in the development of wound healing but also contributes to pathological conditions, such as cancer and fibrosis [85]. Historically, the functional role of EMT in pathological conditions has been classified into type Ι EMT (embryogenesis), type II EMT (pathological fibrosis), and type III EMT (tumorigenesis) [67,86]. However, subsequent studies using genetically engineered knock-out *mouse* models coupled with lineage tracing strategies rebutted that instead of transdifferentiation into myofibroblasts; the epithelial cells still reside within the epithelial basement membrane in a partial EMT state during EMT but with impaired epithelial function and regenerative potential [84]. *Hsa_circ_0044226* was found to be significantly upregulated in pulmonary samples from IPF patients, compared with healthy controls. Functional experiments suggested that *hsa_circ_0044226* plays a regulatory role in EMT of PF in vitro and in vivo, and the downregulation of *hsa_circ_0044226* using a targeted shRNA alleviated both the transforming growth factor-β1 (TGF-β1)-induced fibrosis in RLE-6TN cells and bleomycin-induced mouse model of IPF through the inhibition of EMT. This protective effect was mediated by decreased expression of CDC27, the overexpression of which can reverse the antifibrotic effect of *hsa_circ_0044226* knockdown through activation of EMT. Taken together, downregulation of *hsa_circ_0044226* exerted antifibrotic effects in PF by suppressing EMT through inhibiting CDC27 [66]. Another study identified that the *circular CDR1as*/*miR-7*/TGFBR2 axis exerts a moderating effect on EMT in silica-induced PF. *CircRNA CDR1as* was upregulated in silica-treated pulmonary epithelial cells. Acting as miRNA sponges, *circRNA CDR1as* could sponge *miR-7* to release its target gene transforming growth factor beta receptor 2 (TGFBR2), thereby promoting the EMT process during PF [72]. Similarly, *circular ZC3H4 RNA (circZC3H4)* serves as a *miR-212* sponge to regulate the expression of Zinc finger CCCH-type containing 4 protein (ZC3H4), which was found to be involved in silica-induced EMT, as well as its accompanying migratory characteristics. Jiang et al. reported that the expression of *circZC3H4* was increased in epithelial cells exposed to SiO_2_. Subsequent CRISPR/Cas9 system and siRNAs techniques revealed that the *circZC3H4*/*miR-212*/ZC3H4 axis regulated the EMT process via endoplasmic reticulum stress (ERS) but not autophagy during PF [71]. A recent study investigated the antifibrotic effects of atractylon on EMT of pulmonary epithelial cells in asthma-induced PF. Mechanistic experiments suggested that atractylon treatment downregulated ovalbumin (OVA)-induced *circRNA-0000981* and TGFBR2 expression but upregulated the expression of *miR-211-5p*. Luciferase reporter assay validated that the downregulation of circRNA-0000981 expression suppressed TGFBR2 by sponging *miR-211-5p*, suggesting the protective effect of atractylon may be mediated by the *mmu_circ_0000981*/*miR-211-5p*/TGFBR2 axis. In vivo studies further demonstrated that atractylon treatment suppressed PF in OVA-induced asthma by attenuating EMT via *circRNA-0000981*/*miR-211-5p*/TGFBR2 [73]. Taken together, these studies demonstrated that the interaction between circRNA and miRNA may exert important functions and provide potential therapeutic targets in pulmonary fibrosis.

### 4.3. Macrophage Activation and Polarization

CircRNAs have also been studied in the context of MA and polarization. Given the critical role of the immune dysregulation in the pathophysiology of PF, multiple studies of individuals with PF have focused on monocyte-derived alveolar macrophages and M2 macrophages, which contribute to the fibrotic niche [4]. Following tissue injury, monocytes are recruited and differentiated into M1 phenotype (classical activation, proinflammatory) and M2 phenotype (alternative activation, profibrotic), with a predominance of M2 macrophages [87]. The activation and polarization of macrophages promote fibrosis through multiple mechanisms, including fibroblast activation, myofibroblast differentiation, and ECM remodeling [87]. To determine whether circRNAs are involved in the pathophysiologic process of silicosis, Zhou et al. investigated the molecular mechanisms and functional effects of circRNAs on MA. CircRNA microarray analysis in a *murine* model of SiO_2_-induced silicosis showed that the expression level of *circHECTD1* was decreased in lung tissues from SiO_2_-treated mice, which was further verified in macrophages exposed to SiO_2_ by qRT-PCR analysis and fluorescence in situ hybridization (FISH) assay. Transfection with the *circHECTD1* lentivirus mitigated SiO_2_-induced phenotypic transformation of the macrophages, as well as cell migration, confirming the involvement of *circHECTD1* in SiO_2_-induced MA. Results from RNA-binding protein immunoprecipitation assay demonstrated that *circHECTD1* interacts with HECTD1 and negatively regulated the protein expression of HECTD1, possibly by competing with the pre-mRNA splicing machinery to inhibit host gene expression. Moreover, tissue samples from silicosis patients also confirmed the upregulation of HECTD1. Further gain- and loss-of-function experiments revealed that HECTD1 mediates MA via ZC3H12A ubiquitination, and HECTD1 in macrophages is also involved in fibroblast activation and migration. Collectively, SiO_2_-activated macrophages promoted fibroblast proliferation and migration via the *circHECTD1*/HECTD1 pathway [65]. Apart from EMT, the *circular RNA ZC3H4 (circZC3H4)*/*miR-212*/ZC3H4 pathway was also reported to be involved in MA in the *murine* model of silicosis and the primary cultures of alveolar macrophages from patients, as well as the RAW264.7 macrophage cell line. On the other hand, SiO_2_ concurrently induces MA and the upregulation of ZC3H4 protein, while the knockdown of ZC3H4 by the CRISPR/Cas9 system rescued the phenotypic transformation and decreased cell viability of macrophages induced by SiO_2_; on the other hand, results from the high-throughput screening demonstrated that the expression of circZC3H4 was increased in *mice* subjected to SiO_2_ treatment, which was further confirmed by in situ hybridization. The regulatory pathway of *circZC3H4*/*miR-212*/ZC3H4 was further predicted using bioinformatics analysis and confirmed by RNA in situ hybridization assays, RNA pulldown assay, and base pair mutations of *circZC3H4* [70]. Together, these data support the idea that targeting the circRNA/miRNA/protein axis may provide novel therapeutic strategies for MA and polarization in PF.

### 4.4. Fibroblast Activation and Fibroblast-To-Myofibroblast Transition

During PF, fibroblasts are activated and become myofibroblasts, being the major source of ECM [88]. Many individual circRNAs have been shown to function by activating fibroblast [61,68,74,76,77,78,79,89,90]. One example is provided by *circHECTD1*, which was downregulated in HPF-α cells after exposure to silica, as well as in fibroblasts derived from patients with silicosis. The downregulation of *circHECTD1* was associated with an upregulation of HECTD1, which mediates autophagy in fibroblasts exposed to silica, thereby regulating the activation, proliferation, and migration of fibroblast [76]. Similarly, *ciR-012091*, which is also decreased in *murine* model of silicosis, promotes the proliferation and migration of fibroblasts and thereby leads to PF. Functionally, SiO_2_-induced downregulated *ciR-012091* increases its downstream target PPP1R13B, thereby mediating fibroblast proliferation and migration via ERS and autophagy [77]. In addition, Li et al. reported that both *circ949* and *circ057* share a common direct targeted miRNA *miR-29b-2-5p*. Further gain- and loss-of-function experiments showed that *miR-29b-2-5p* blocked the proliferation and migration of fibroblasts [61]. *Circular RNA TADA2A (circTADA2A)* is another example of a circRNA that was downregulated in both IPF primary human lung fibroblasts and human IPF fibroblastic cell lines. Mechanistically, *circTADA2A* repressed the activation and proliferation of fibroblasts induced by several fibrogenic growth factors. Results of FISH combined with luciferase reporter assays and RNA pull-down confirmed that *circTADA2A* function as sponges of *miR-526b* and *miR-203*, thereby releasing the expression of Caveolin (Cav)-1 and Cav-2. Specifically, *circTADA2A* inhibits the excessive deposition of ECM and attenuates PF by suppressing lung fibroblasts activation via *miR-526b*/Cav1 and reducing lung fibroblasts proliferation via *miR-203*/Cav2 [78]. On the contrary, the expression of *circular RNA HIPK3 (circHIPK3)* was increased in vitro experiment of TGF-β1-induced lung fibroblasts and in vivo experiment of fibrotic *murine* lung tissues induced by silica, while inhibition of *circHIPK3* attenuated the activation, proliferation, and glycolysis of fibroblasts. Functioning as a sponge of *miR-30a-3p*, *circHIPK3* negatively regulates its expression and enhances the expression of FOXK2, which is the direct target of *miR-30a-3p* and a driver transcription factor of glycolysis. Accordingly, *miR-30a-3p* overexpression or FOXK2 knockdown blocked fibroblast activation induced by TGF-β1 and abrogated the profibrotic effects of *circHIPK3*. Adeno-associated virus (AAV)-mediated *circHIPK3* silence or *miR-30a-3p* overexpression alleviated silica-induced PF in vivo. Therefore, *circHIPK3* could enhance the expression of FOXK2 via sponging *miR-30a-3p*, thereby facilitating fibroblast glycolysis and activation in PF [74]. *Circular RNA HIPK2 (circHIPK2)* was also upregulated in human pulmonary fibroblasts exposed to SiO_2_. Functional experiments and specific knockdown of *circHIPK2* with siRNA confirmed that *circHIPK2* takes part in SiO_2_-induced ERS via promoting the expression of sigma-1 receptor, which is an endoplasmic reticulum chaperone. As ERS further enhances the activation, proliferation, and migration of fibroblasts, *circHIPK2* may play a profibrotic role in silicosis [79]. In addition to chemical stimulation and metabolic alteration, changes in mechanical properties play a substantial role in regulating lung fibrogenesis [89,90]. A recent study reported that a novel *circRNA ANKRD42 (circANKRD42)*, which is from peripheral blood of patients with IPF, participated in PF through mediating the crosstalk between mechanical stiffness and biochemical signals and facilitating FMT. Mechanistic studies revealed that the *circANKRD42* biogenesis was activated by the upstream hnRNP L via promoting RNA reverse splicing. As *circANKRD42* simultaneously served as a sponge of *miR-324-5p* and *miR-136-5p*, two signal pathways mediating the profibrotic effect of *circANKRD42* have been uncovered. First, the *circANKRD42* sponged *miR-324-5p* to promote the AJUBA expression, which blocked the binding between phosphorylated yes-associated protein 1 (YAP1) and large tumor suppressor kinase 1/2, leading to increased YAP1 entering the nucleus. Second, *circANKRD42* also sponged *miR-136-5p* to promote the YAP1 translation. Then the accumulating YAP1 in nucleus bound to TEAD, which initiated the transcription of genes related to mechanical stiffness, such as F-actin and Myo1c, thus worsening PF [68].

Myofibroblasts are the dominant effector cells in IPF, which deposit collagen and localize to fibrotic foci. Among multiple cellular sources from which that myofibroblast could originate, resident fibroblasts via FMT are considered to be the major contributors to myofibroblasts in the lung [4]. A few circRNAs have been proposed to influence the process of FMT, including *circHIPK3* and *circ0044226* [75,91]. *CircHIPK3* was reported to be upregulated both in the *murine* model of BIPF, FMT-derived myofibroblasts, and clinical samples of patients with idiopathic PF. Notably, silencing *circHIPK3* by AAV shRNAs markedly ameliorates FMT and collagen deposition both in vivo and in vitro. Mechanistically, *circHIPK3* functions as an endogenous *miR-338-3p* sponge and inhibits *miR-338-3p* activity, whose inhibition increased the expression of SOX4 and COL1A1, thereby promoting FMT and PF [91]. *Circ0044226* (named *circRNA_102100*) is another circRNA that was found to be increased in *murine* model of BIPF and FMT-derived myofibroblasts, which was distributed mostly in the cytoplasm. In vivo combined with in vitro experiments demonstrated that silencing *circ0044226* by siRNA transfection inhibits fibroblast proliferation and differentiation, suggesting the regulatory role of *circ0044226* in FMT. Dual-luciferase reporter assay and FISH assay further confirmed that *circ0044226* function as an *miR-7* sponge and *miR-7* represses sp1 mRNA via pairing to its 3′UTR. Thereby, intervention of the *circ0044226/miR-7*/sp1 network may represent a promising therapy for PF [75].

In addition, aberrant cross-talk of epithelium-fibroblasts is also involved in FMT and PF [92,93]. Recently, Bai and colleagues reported that *circRNA_0026344* mediates the CS-induced aberrant crosstalk of epithelium-fibroblasts in smoking-induced PF. By using microarray analysis and subsequent qRT-PCR, they verified that *circRNA_0026344* was substantially decreased following cigarette smoke extract (CSE) exposure in human bronchial epithelial cells (HBE cells). Then the regulatory relationship between *circRNA_0026344* and *miR-21* was speculated by competing endogenous RNA (ceRNA) network analysis and verified by RNA pull down assays. Functional analysis by qRT-PCR combined with the transfection of high expression plasmid revealed that the downregulated *circRNA_0026344* by CSE increased *miR-21* levels by acting as a miRNA sponge for *miR-21* and reducing its adsorption in airway epithelial cells. The exosomal *miR-21* from CSE-treated epithelial cells bound to the 3′UTR region of Smad7 mRNA decreases Smad7 and subsequently activates the TGF-β1/Smad3 pathway in fibroblasts, which further promotes fibroblasts to differentiate into myofibroblasts [80].

### 4.5. Other Biological Processes

Several lines of evidence indicated that injury and dysfunction of the lung epithelium is central to initiating the pathogenic process of PF [93,94]. Airway inflammation is the pathological basis of many fine particulate matter (PM2.5)-induced pulmonary diseases, including PF [95]. *Circular RNA 406961 (circ_406961)* has been shown to be decreased in human bronchial epithelial cells (BEAS-2B) exposed to PM2.5. This circRNA was proposed to inhibit PM2.5-induced inflammatory reaction. Functional experiments further confirmed that *circ_406961* interacted with interleukin enhancer-binding factor 2 (ILF2), leading to the inhibition of signal transducer and activator of transcription 3 (STAT3) and mitogen-activated protein kinase 8 (MAPK8, JNK) pathways and subsequent release of the inflammatory factors IL-6 and IL-8 [81].

## 5. Diagnostic and Therapeutic Potential of Targeting circRNAs in Pulmonary Fibrosis

Accurate diagnosis and treatment of specific forms of ILD from IPF are frequently challenging [96]. To establish a definitive diagnosis with prognostic significance, clinical and radiological data combined with invasive examinations, such as bronchoalveolar lavage, lung biopsy, and transbronchial lung cryobiopsy, are required [97,98]. In addition, progressive fibrosis is not exclusive to the UIP histological pattern of high-resolution CT [99]. Given that (1) circRNAs are stable and resistant to RNA exonuclease or RNase R, (2) the expression pattern of circRNAs is tissue- and cell-type-specific, and (3) the abundance of circ-RNAs in body fluids are detectable for non-invasive detection, circRNAs may potentially serve as diagnostic biomarkers and therapeutic targets of PF. For example, *hsa_circ_0058493* from peripheral blood lymphocyte was highly expressed in both silicosis and IPF cases, which was identified by whole transcriptome RNA-seq strategy and verified by multi-stage validation [64]. In addition, the results of RNA FISH assay and qRT-PCR demonstrated that *circHIPK3* is upregulated in the lung tissues of IPF patients, compared with healthy people [91]. Furthermore, serum levels of *circANKRD42* were higher in patients with IPF (*n* = 25) [68]. However, these potential biomarkers have not been rigorously tested for the correlation with clinicopathological manifestations, as well as severity or prognosis of PF. Taken together, these data suggest that aberrant circRNAs from blood samples are strongly associated with PF. The investigation of the circulating circRNA profile in PF for non-invasive approaches for diagnosis and prognosis would be critical areas of future research.

Given that circRNAs are critical in many cellular processes of PF through diverse modes of action, therapy targeting circRNAs may be feasible and promising. Although still in its infancy, recent attempts to engineer the circRNA-based therapy in different contexts have provided new insights into RNA therapy [100]. By silencing or overexpression of circRNA, it is conceivable that they could modulate the pathophysiological processes of PF, thereby alleviating or reversing the disease progression. For antifibrotic circRNAs, the overexpression of circRNAs could be performed by the delivery system of lentivirus or AAV vectors and extracellular vesicles or plasmid conjugated with colloidal gold nanoparticles [101]. On the contrary, for profibrotic circRNAs, the silencing of the circRNAs is typically achieved using RNA interference approach or the CRISPR-Cas genome editing system [102,103,104]. However, challenges, such as off-target effects and immunogenicity, still remain before translation from bench to bedside. So far, circRNA-based therapeutic approaches have not yet been shown to outperform the current treatment of PF, and further investigation of their potential for RNA therapy is warranted.

## 6. Conclusions and Future Perspectives

Over the last decades, the advent of high-throughput RNA-seq technologies, combined with dedicated bioinformatics algorithms for circRNA annotation have enabled a better understanding of the characterization, biological functions, and pre-clinical applications of circRNAs. As a relatively young field of research, circRNA holds great translational promise. In this regard, well-designed cohort studies with large samples and internal and external validation are critical in the future. However, several questions and challenges in the field need to be addressed. Firstly, the majority of currently available evidence linking circRNAs and PF is derived from in vitro experiments or pre-clinical models. Indeed, the important roles of circRNA should also be verified during human pathology, particularly in the relation to clinicopathological and survival parameters, which is of paramount importance for the clinical translation of promising biomarkers. Secondly, studies elaborating the expression pattern at the single-cell level are still lacking. As the expression pattern of certain circRNA may be cell-dependent, for example, *circHECTD1* [65,76,82], future endeavors should, therefore, be made to characterize the spatial resolution and temporal expression pattern in specific cell types. Thirdly, apart from miRNA sponge mechanisms, other mechanisms of circRNAs in PF have rarely been studied. Similarly, research on epigenetic modifications, such as circRNA methylation in PF are understudied [105,106]. Future work is needed to fill these knowledge gaps, as well as to translate from bench to bedside.

## Figures and Tables

**Figure 1 ijms-23-10493-f001:**
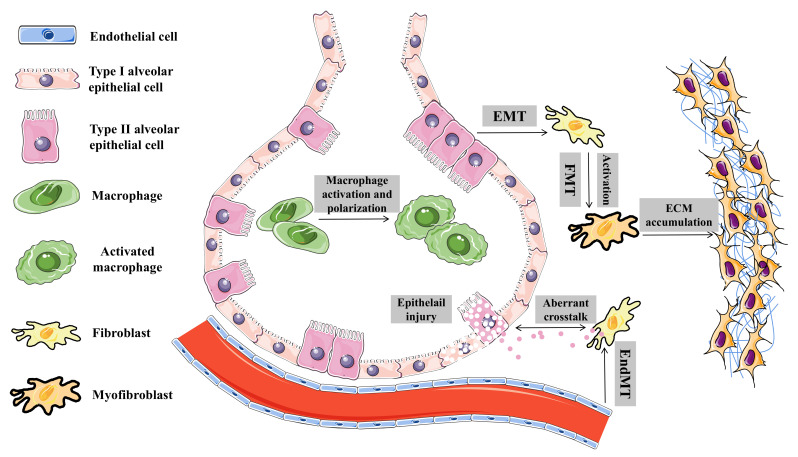
Pathophysiology of pulmonary fibrosis. Underlying pathophysiology of pulmonary fibrosis involves repetitive epithelial injury, epithelial-to-mesenchymal transition, macrophage activation, polarization, persistent fibroblast activation, and fibroblast-to-myofibroblast transition and subsequent excessive deposition of extracellular matrix.

**Figure 2 ijms-23-10493-f002:**
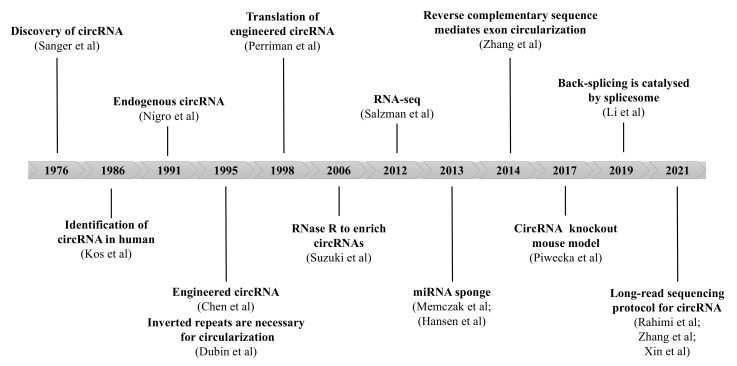
Overview of research history of circRNA. The discovery of circRNA starts with identification of viroids, which was published in 1976. Subsequent investigations and novel technologies revealed the biogenesis and properties of circRNA. (see Refs. [10,11,12,13,14,15,16,17,18,19,20,21,22,23,24,25]).

**Figure 3 ijms-23-10493-f003:**
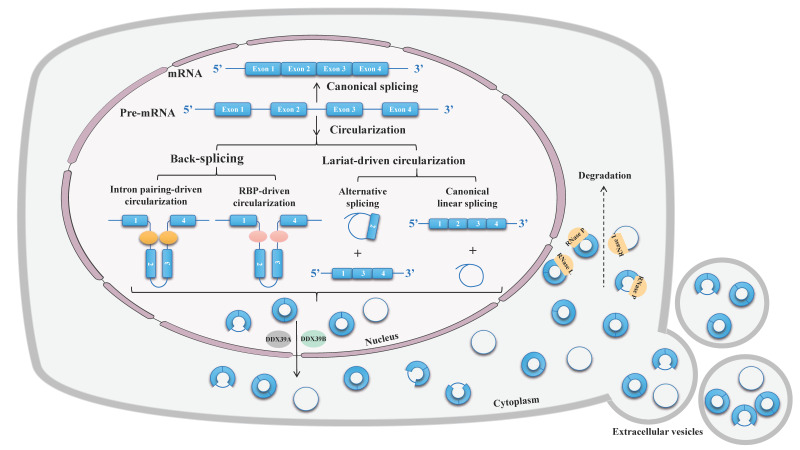
Biogenesis, trafficking, and degradation of circRNA. Different from the canonical splicing that joins an upstream 5′ splice site with a downstream 3′ splice site; circRNA are produced from back-splicing of precursor mRNA, in which a downstream 5′ splice site joins an upstream 3′ splice site in reverse order. In addition, intron lariats from conventional splicing can retain a circular form when escape debranching. Apart from biogenesis, trafficking and degradation also regulate the abundance of circRNA. Nuclear export of circRNAs to the cytoplasm occurs in a length-dependent manner and requires multiple proteins, such as spliceosome RNA helicase DDX39B (for long circRNAs) and ATP-dependent RNA helicase DDX39A (for short circRNAs). CircRNA also can be degraded by endonucleases or further exported to extracellular vesicles.

**Figure 4 ijms-23-10493-f004:**
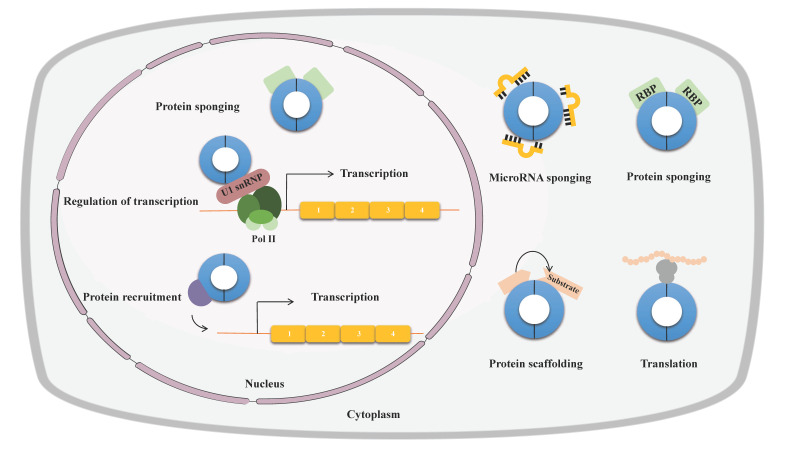
Biological functions of circRNA. CircRNAs function as important gene regulators through multiple actions, including miRNA or protein sponges or decoys, templates for translation, protein scaffolding, and recruitment or enhancer of protein function.

**Figure 5 ijms-23-10493-f005:**
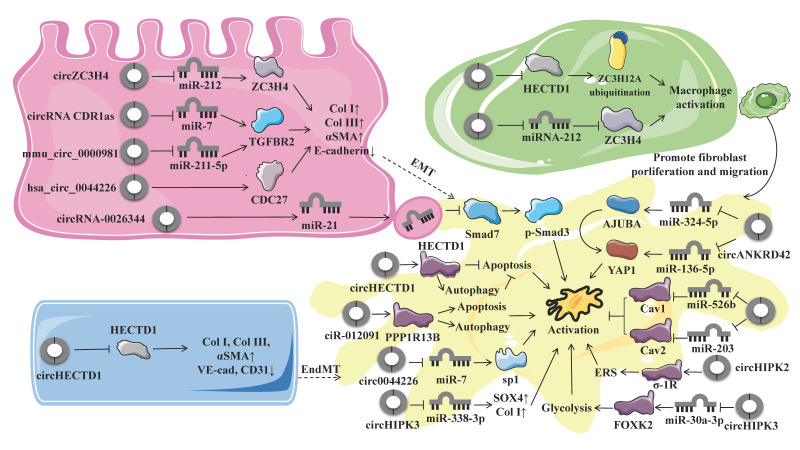
circRNAs in pulmonary fibrosis and their influence on biological processes. CircRNA dysregulation intrigues a complex interplay of signaling pathways in endothelial-to-mesenchymal transition, epithelial-to-mesenchymal transition, macrophage activation, and fibroblast-to-myofibroblast transition.

**Table 1 ijms-23-10493-t001:** circRNAs expression profiles in pulmonary fibrosis.

Species	Model	Method	Criteria	Total circRNA	Upregulated	Downregulated	Reference
Mice	Mouse model of SiO_2_-induced silicosis	Microarray analysis	Fold change > 2.0; *p* < 0.05	120	73	47	[65]
Mice	Mice model of BIPF	RNA sequencing	Fold-change ≥ 3.0	74	-	-	[61]
Human	IPF Patients (GEO database [GSE102660])	Bioinformatics analysis	|Fold change| > 1.5; −log10 *p*-value > 1.3	45	22	23	[67]
Human	Silicosis patients	RNA sequencing	Fold change > 1.5; *p* < 0.05	243	139	104	[64]
Human	IPF Patients (GEO database [GSE102660])	Bioinformatics analysis	*p* < 0.05	316	200	116	
Human	Peripheral blood samples of IPF patients	Microarray hybridization	|Fold change| ≥ 1.5; *p* ≤ 0.05	67	38	29	[60,68]
Rats	Rat model of BIPF	RNA sequencing	Fold change > 2.0; *p* < 0.05	10	2	8	[62]
Rats	Rat model of BIPF	Whole transcriptome sequencing	|Fold change| > 2; *p* ≤ 0.05	605	287	318	[63]

Abbreviations: SiO_2_, silicon dioxide; IPF, idiopathic pulmonary fibrosis; BIPF, bleomycin-induced pulmonary fibrosis.

**Table 2 ijms-23-10493-t002:** Dysregulated circRNAs in pulmonary fibrosis.

CircRNA	Dysregulation	Role in PF	Target Gene; Related Molecular	Function	Model	Cell	Reference
Endothelial–mesenchymal transition							
circHECTD1	Up	Pro	HECTD1	EndMT	Murine model of silicosis	MML1	[69]
Macrophage activation							
circHECTD1	Down	Anti	HECTD1; ZC3H12A	MA	Murine model of silicosis	RAW264.7	[65]
circZC3H4	Up	Pro	ZC3H4	MA	Silicosis patients	RAW264.7	[70]
Epithelial–mesenchymal transition							
hsa_circ_0044226	Up	Pro	CDC27	EMT	Murine model of BIPF	RLE-6TN	[67]
circZC3H4	Up	Pro	miR-212; ZC3H4	EMT	Murine model of silicosis	MLE-12, A549, BEAS-2B	[71]
ciRS-7	Up	Pro	miR-7; TGFBR2	EMT	Murine model of silicosis	HBE, A549; MRC-5, NIH/3T3	[72]
mmu_circ_0000981	Up	Pro	miR-211-5p; TGFBR2	EMT	Murine model of asthma	TC-1	[73]
Fibroblast-to-myofibroblast transition							
circHIPK3	Up	Pro	miR-338-3p; SOX4 and COL1A1	FMT	Murine model of BIPF	WI-38	[74]
circ0044226	Up	Pro	miR-7; sp1	FMT	Murine model of BIPF	WI-38	[75]
circANKRD42	Up	Pro	miR-324-5p, miR-136-5p; AJUBA, YAP1	FMT	Murine model of BIPF and IPF patients	MRC-5	[68]
Fibroblast activation							
circHECTD1	Down	Anti	HECTD1	FA	Murine model of silicosis	HPF-α	[76]
ciR-012091	Down	Anti	PPP1R13B	FA	Murine model of silicosis	L929 and HPF-α	[77]
circ949 and circ057	Up	Pro	miR-29b-2-5p; STAT3 phosphorylation	FA	Murine model of BIPF	L929	[61]
circTADA2A	Down	Anti	miR-526b, miR-203; Caveolin-1, Caveolin-2	FA	Murine model of BIPF	Fibroblasts	[78]
circHIPK3	Up	Pro	miR-30a-3p; FOXK2	FA	Murine model of silicosis	MRC-5	[74]
circHIPK2	Up	Pro	σ-1R	FA	NA	HPF-α	[79]
circRNA 0026344	Down	Anti	miR-21; Smad7	FA	Murine model of cigarette smoke-induced PF	HBE, MRC-5	[80]
circ_406961	Down	Anti	ILF2; STAT3, MAPK8, JNK	Airway inflammation	NA	BEAS-2B	[81]

Abbreviations: PF, pulmonary fibrosis; EndMT, endothelial–mesenchymal transition; MA, macrophage activation; EMT, epithelial-to-mesenchymal transition; FMT, fibroblast-to-myofibroblast transition; FA, fibroblast activation; BIPF, bleomycin-induced pulmonary fibrosis; IPF, idiopathic pulmonary fibrosis; NA, not applicable.

## Data Availability

Not applicable.

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
