# Peer review of "Role of Circular RNAs in Pulmonary Fibrosis"

_ijms, 2022, doi:10.3390/ijms231810493_

Round 1

Reviewer 1 Report

Dear authors, thanks for this nice and comprehensive review on circulating RNAs in pulmonary fibrosis. 

I have only some small minor comments regarding some typos and misspellings.

Please include in figure 2 a dot after et al. and add the reference number (for example, Sanger et al. Ref: 10)

Line 66: do not separate RNA into two lines

Line 282: cav-1 and cav-2 not cav-1 and cav2

Line 347: "to" is not necessary for this sentence

Line 350: circular needs to start with a capital C

Line 364: do not separate RNA into two lines

Line 370 and 371: please, change "normal" to healthy.

Line 401: human and not huamn

Author Response

Dear Reviewer,

Re: Manuscript reference number ijms-1897986.

Thank you for your letter and for the comments concerning our manuscript entitled “Role of circular RNAs in pulmonary fibrosis” (ijms-1897986). Please find below a revised version with corrections highlighted of our manuscript “Role of circular RNAs in pulmonary fibrosis”, which we would like to resubmit for publication as a research article in International Journal of Molecular Sciences.

Your letter and those comments were highly insightful and enabled us to greatly improve the quality of our manuscript. We have studied comments and requirements carefully and have made correction which we hope meet with approval. We also carefully proof-read the manuscript to minimize typographical, grammatical, and bibliographical errors. In the following pages are our point-by-point responses to each of the reviewer’s comments.

Revisions in the text are shown using underline and red highlight [example] for additions, and deletions are marked up using the “track changes” function. In accordance with reviewers’ comments, we modified inappropriate statements, corrected errors in spelling, syntax, and grammar, added descriptive legends for Figures 1-5 after careful consideration. We hope that the revisions in the manuscript and our accompanying responses will be sufficient to make our manuscript suitable for publication in International Journal of Molecular Sciences.

We shall look forward to hearing from you at your earliest convenience.

Yours sincerely,

Xuehan Li, xuehanli@scu.edu.cn; Tel.: +86 18980099133;
Rurong Wang, wangrurong@scu.edu.cn; Tel.: +86 18980601563. 

Responses to the comments of Reviewer 1

Comments and Suggestions for Authors:

Dear authors, thanks for this nice and comprehensive review on circulating RNAs in pulmonary fibrosis.

Response: Thank you very much for the time and effort that you have put into reviewing our manuscript. Your comments and suggestions have enabled us to significantly improve our work.

Specific comments:

I have only some small minor comments regarding some typos and misspellings.

Response: We apologize for the mistakes in the manuscript and also carefully checked the entire manuscript for typographic,grammatical and formatting errors.

Comment 1: Please include in figure 2 a dot after et al. and add the reference number (for example, Sanger et al. Ref: 10)

Response: We deeply appreciate the reviewer’s suggestion. According to the reviewer’s comment, we include a dot after et al. and add the reference number in Figure 2.

Comment 2: Line 66: do not separate RNA into two lines

Response: We apologize for the mistake and we have corrected it according to your ideas.

Comment 3: Line 282: cav-1 and cav-2 not cav-1 and cav2

Response: We apologize for the mistake and we have corrected it according to your ideas.

Comment 4: Line 347: "to" is not necessary for this sentence

Response: We apologize for the mistake and we have corrected it according to your ideas.

Comment 5: Line 350: circular needs to start with a capital C

Response: We apologize for the mistake and we have corrected it according to your ideas.

Comment 6: Line 364: do not separate RNA into two lines

Response: We apologize for the mistake and we have corrected it according to your ideas.

Comment 7: Line 370 and 371: please, change "normal" to healthy.

Response: We apologize for the mistake and we have corrected it according to your ideas.

Comment 8: Line 401: human and not huamn

Response: We apologize for the mistakes in the manuscript and also carefully checked the entire manuscript for typographic,grammatical and formatting errors.

We have tried our best to improve the manuscript and made some changes in the manuscript. These changes will not influence the content and framework of the paper.

We appreciate for Editors/Reviewers’ warm work earnestly, and hope that the correction will meet with approval. Once again, thank you very much for your comments and suggestions.

Reviewer 2 Report

This review on the involvement of circular RNA in pulmonary fibrosis is well and clearly structured. The authors compile the latest publications in this relatively new field and provide arguments to consider circRNAs as diagnostic tools and potential therapeutic targets in pulmonary fibrosis.

I would like to make some recommendations to the authors that could improve the quality of the manuscript:

Although the Figures are nice and informative, a descriptive legend for each of them would be advisable. Especially for Figure 5, which is highly complex.

Some phrases in the text are meaningless due to the use of certain words or the absence of them:

Page 3, around line 71

…in many types of PF, including silicosis, idiopathic PF, cigarette smoke-induced PF, in which circRNA dysregulation intrigues a complex interplay of cell types and signaling pathways, such as endothelial-mesenchymal transition (EndMT),…

Perhaps the authors wanted to use "involves" or “leads to” or “gives rise to”, instead of "intrigues"

Page 9, lines 220-221

Taken together, these studies demonstrated that the interaction between circRNA and miRNA may exert important functions and provide potential therapeutic targets in pulmonary.

It seems that a last word is needed after "pulmonary" at the end of the sentence ("pulmonary fibrosis"?).

Pages 10-11, lines 296-298

Functional experiments and specific knockdown of circHIPK2 with siRNA confirmed that circHIPK2 take part in SiO2-induced ERS via promoting the expression of […..], which is an endoplasmic reticulum chaperone.

Missing name of the chaperone.

 Page 12, lines 350-352

circular RNA 406961 (circ_406961) has been shown to be decreased in human bronchial epithelial cells (BEAS-2B) were exposed to PM2.5 and inhibits PM2.5-induced inflammatory reaction.

Rephrase to make it understandable.

I have found some typographical errors throughout the text, so a final spell check of the text would be advisable. For example:

Page 12, lines 397-398

In this regard, well-design [well-designed?] cohort studies with large samples and nternal and external validation are critical in the future.

Page 13, lines 401-402

Indeed, the importance roles of circRNA should also be verified during huamn pathology, …

Author Response

Dear Reviewer,

Re: Manuscript reference number ijms-1897986.

Thank you for your letter and for the comments concerning our manuscript entitled “Role of circular RNAs in pulmonary fibrosis” (ijms-1897986). Please find below a revised version with corrections highlighted of our manuscript “Role of circular RNAs in pulmonary fibrosis”, which we would like to resubmit for publication as a research article in International Journal of Molecular Sciences.

Your letter and those comments were highly insightful and enabled us to greatly improve the quality of our manuscript. We have studied comments and requirements carefully and have made correction which we hope meet with approval. We also carefully proof-read the manuscript to minimize typographical, grammatical, and bibliographical errors. In the following pages are our point-by-point responses to each of the reviewer’s comments.

Revisions in the text are shown using underline and red highlight [example] for additions, and deletions are marked up using the “track changes” function. In accordance with reviewer’s comments, we modified inappropriate statements, corrected errors in spelling, syntax, and grammar, and added descriptive legends for Figures 1-5 after careful consideration. We hope that the revisions in the manuscript and our accompanying responses will be sufficient to make our manuscript suitable for publication in International Journal of Molecular Sciences.

We shall look forward to hearing from you at your earliest convenience.

Yours sincerely,

Xuehan Li, xuehanli@scu.edu.cn; Tel.: +86 18980099133;
Rurong Wang, wangrurong@scu.edu.cn; Tel.: +86 18980601563. 

Responses to the comments of Reviewer 2

Comments and Suggestions for Authors:

This review on the involvement of circular RNA in pulmonary fibrosis is well and clearly structured. The authors compile the latest publications in this relatively new field and provide arguments to consider circRNAs as diagnostic tools and potential therapeutic targets in pulmonary fibrosis.

Response: Thank you very much for the time and effort that you have put into reviewing our manuscript. Your comments and suggestions have enabled us to significantly improve our work.

Specific comments:

I would like to make some recommendations to the authors that could improve the quality of the manuscript:

Comment 1: Although the Figures are nice and informative, a descriptive legend for each of them would be advisable. Especially for Figure 5, which is highly complex.

Response: We deeply appreciate the reviewer’s suggestion. According to the reviewer’s comment, we added descriptive legends for Figures 1-5. Thanks again for your professional advice.

Comment 2: Some phrases in the text are meaningless due to the use of certain words or the absence of them:

Page 3, around line 71

…in many types of PF, including silicosis, idiopathic PF, cigarette smoke-induced PF, in which circRNA dysregulation intrigues a complex interplay of cell types and signaling pathways, such as endothelial-mesenchymal transition (EndMT),…

Perhaps the authors wanted to use "involves" or “leads to” or “gives rise to”, instead of "intrigues"

Response: We appreciate it very much for this insightful suggestion, and we have corrected it according to your ideas.

Comment 3: Page 9, lines 220-221

Taken together, these studies demonstrated that the interaction between circRNA and miRNA may exert important functions and provide potential therapeutic targets in pulmonary.

It seems that a last word is needed after "pulmonary" at the end of the sentence ("pulmonary fibrosis"?).

Response: We apologize for the mistake and we have corrected it according to your ideas.

Comment 4: Pages 10-11, lines 296-298

Functional experiments and specific knockdown of circHIPK2 with siRNA confirmed that circHIPK2 take part in SiO2-induced ERS via promoting the expression of […..], which is an endoplasmic reticulum chaperone.

Missing name of the chaperone.

Response: We apologize for the mistake and we have added the missing name of the chaperone.

Comment 5: Page 12, lines 350-352

circular RNA 406961 (circ_406961) has been shown to be decreased in human bronchial epithelial cells (BEAS-2B) were exposed to PM2.5 and inhibits PM2.5-induced inflammatory reaction.

Rephrase to make it understandable.

Response: We are extremely grateful to Reviewer for reviewing the paper so carefully and pointing out this problem. We have rethought the argument in depth and adjusted the text in the manuscript.

Comment 6: I have found some typographical errors throughout the text, so a final spell check of the text would be advisable. For example:

Page 12, lines 397-398

In this regard, well-design [well-designed?] cohort studies with large samples and nternal and external validation are critical in the future.

Response: We apologize for the mistakes in the manuscript and also carefully checked the entire manuscript for typographic,grammatical and formatting errors.

Comment 7: Page 13, lines 401-402

… Indeed, the importance roles of circRNA should also be verified during huamn pathology, …

Response: We apologize for the mistakes in the manuscript and also carefully checked the entire manuscript for typographic,grammatical and formatting errors.

We have tried our best to improve the manuscript and made some changes in the manuscript. These changes will not influence the content and framework of the paper.

We appreciate for Editors/Reviewers’ warm work earnestly, and hope that the correction will meet with approval. Once again, thank you very much for your comments and suggestions.

Reviewer 3 Report

This paper aims to present the current knowledge and understanding of circular RNAs in the field of pulmonary fibrosis. The paper is well written, easy to read, include a number of recent papers in the area, and is for the interest of readers from the biomedical discipline, more precisely, for those interested in pulmonary diseases. However, this referee believes that sometimes it is lacking a bit of depth. It reads more like a listing of papers and ideas, rather than drawing them together to tell a story (for example the part of “Biological roles and regulatory mechanisms of circRNAs in pulmonary fibrosis” ).

The review briefly describes the concept and properties of circular RNAs, focusing in the biological functions that those molecules exert in vivo, and the molecular mechanism involved.

In my opinion, this work could be improved with some changes.

-       There is a lack of information in the figure caption (specially in figures 1,3, 4 and 5). The images should be self-descriptive. The images are OK, but the text of the figure caption should be more informative and a description of the figures and the components of the figures would be strongly recommended (i,e, what is DDX39A in figure 3. It is explained in the text, but not in the figure caption).

-       Authors could organize and list the papers of Tables 1 and 2 following a general criteria (i.e. “species employed” in Table 1 or “animal model” in Table 2), instead of listing a number of papers.

-       In the paragraph of page 5 lines 131-145, authors list a number of studies in which they mention that “a number” of circRNAs where upregulated or downregulated, but they do not analyze if they are a specific circRNA that it is always upregulated or downregulated in all the models, or if a specific family of orthologous genes circRNA plays the same role.

-       Page 3 lines 69 and 70. Please, consider to include the description of the different types of PF in the introduction dedicated to “Pulmonary fibrosis” (First paragraph in page 1).

-       Page 4 lines 105. When speaking about “making them exceptionally stable once formed “, general readers would appreciate to include the half life of those molecules.

-       Page 8 line 169. Authors should consider to employ the term “prevented” instead of  “reversed”.

-       Page 9 line 235. Authors employ , “circular RNA HECT domain E3 ubiquitin protein ligase 1 (circHECTD1))”. They should employ the same criteria when they mention the rest of circRNAs.

-       Page 8 line 198. Authors should consider to employ the term “prevented” instead of  “reversed”.

-       Page 10 line 264. “Many individual circRNAs have been shown to function by activating fibroblast”. Please, provide a reference/s for this statement.

-       Page 12 line 370. “qRT-PCR demonstrated that circHIPK3 is upregulated in the lung tissues of normal people and IPF patients compared with normal people”. Please, check the groups involved in the comparison.

-       Page 11 line 319. A few circRNAs have been proposed to influence the process of FMT, including circHIPK3 and circ0044226. Please, provide a reference/s for this statement.

There is a number of mistakes in the text:

-       Page 2 lines 55 to 57 : “CircRNAs are reported to be implicated in the regulation of many different cellular and pathophysiological processes through diverse mechanisms of action”. This phrase is too vague.

-       Page 4 lines 118. When authors mention that “some nuclear circRNAs can also regulate gene expression by enhancing the of RNA polymerase II”. They are referring to an increase in the transcription of the gene of RNA polymerase, the protein levels of RNA polymerase, or the activity of the protein?. The phrase is incomplete or confusing.

-       Page 4 line 109. There are two “most” in the same phrase. Authors should consider to employ other term/synonym.

-       Page 4 line 112. “Secondly, Some circRNAs”. Some, should be “some” (lowercase letters)

-       Page 5 line 133.” Similarly, A total of 10 circRNAs” . A, should be “a” (lowercase letters)

-       Page 8 line 161. The abbreviation “EndMT” is previously described in page 3 line 72.

-       Page 8 line 179. The term “epthelial” should be “epithelial”, and the abbreviation is previously described in page 1 line 32.

-       Page 9 line 196. The term TGF-β should be described for the first time.

-       Page 8 line 203. The term “upregualted” should be “upregulated”.

-       Page 9 line 205. Authors employ , “circular ZC3H4 RNA (circZC3H4)”. They should employ the same criteria when they mention the rest of circRNAs.

-       Page 9 lines 220-222. . Taken together, these studies demonstrated that the interaction between circRNA and miRNA may exert important functions and provide potential therapeutic targets in pulmonary”. The phrase is incomplete or confusing.

-       Page 9 line 245. “Further Gain- and”. Gain, should be “gain” (lowercase letters)

-       Page 10 line 261. “macrophage activation”….the abbreviation is previously described.

-       Page 10 line 286. “silica., while...” Please, check the punctuation mark.

-       Page 10 line 295. circHIPK2 should be “CircHIPK2” (uppercase letters).

-       Page 11 line 298. “promoting the expression of , which is” . The phrase is incomplete.

-       Page 11 line 337.HBE cells. The source of the cell type should be described.

-       Page 11 line 347. “Several lines of evidence indicated that injury to, and dysfunction”. The phrase is incomplete or confusing.

-       Page 12 line 350. circular RNA 406961 (circ_406961) has been shown to be decreased in human bronchial epithelial cells (BEAS-2B) were exposed to PM2.5 and inhibits PM2.5-induced inflammatory reaction. The phrase is incomplete or confusing.

-       Page 12 line 396. Nternal should be “internal”.

-       Page 13 line 401. Huamn should be “human”.

-       Page 13 line 404. Studies should be “studies” (lowercase letters).

Author Response

Dear Reviewer,

Re: Manuscript reference number ijms-1897986.

Thank you for your letter and for the comments concerning our manuscript entitled “Role of circular RNAs in pulmonary fibrosis” (ijms-1897986). Please find below a revised version with corrections highlighted of our manuscript “Role of circular RNAs in pulmonary fibrosis”, which we would like to resubmit for publication as a research article in International Journal of Molecular Sciences.

Your letter and those comments were highly insightful and enabled us to greatly improve the quality of our manuscript. We have studied comments and requirements carefully and have made correction which we hope meet with approval. We also carefully proof-read the manuscript to minimize typographical, grammatical, and bibliographical errors. In the following pages are our point-by-point responses to each of the reviewer’s comments.

Revisions in the text are shown using underline and red highlight [example] for additions, and deletions are marked up using the “track changes” function. In accordance with reviewer’s comments, we modified inappropriate statements, corrected errors in spelling, syntax, and grammar, and added descriptive legends for Figures 1-5 after careful consideration. We hope that the revisions in the manuscript and our accompanying responses will be sufficient to make our manuscript suitable for publication in International Journal of Molecular Sciences.

We shall look forward to hearing from you at your earliest convenience.

Yours sincerely,

Xuehan Li, xuehanli@scu.edu.cn; Tel.: +86 18980099133;
Rurong Wang, wangrurong@scu.edu.cn; Tel.: +86 18980601563. 

Responses to the comments of Reviewer 3

Comments and Suggestions for Authors:

This paper aims to present the current knowledge and understanding of circular RNAs in the field of pulmonary fibrosis. The paper is well written, easy to read, include a number of recent papers in the area, and is for the interest of readers from the biomedical discipline, more precisely, for those interested in pulmonary diseases. However, this referee believes that sometimes it is lacking a bit of depth. It reads more like a listing of papers and ideas, rather than drawing them together to tell a story (for example the part of “Biological roles and regulatory mechanisms of circRNAs in pulmonary fibrosis” ).

The review briefly describes the concept and properties of circular RNAs, focusing in the biological functions that those molecules exert in vivo, and the molecular mechanism involved.

Response: Thank you very much for the time and effort that you have put into reviewing our manuscript. Your comments and suggestions have enabled us to significantly improve our work.

Specific comments:

In my opinion, this work could be improved with some changes.

Comment 1: There is a lack of information in the figure caption (specially in figures 1,3, 4 and 5). The images should be self-descriptive. The images are OK, but the text of the figure caption should be more informative and a description of the figures and the components of the figures would be strongly recommended (i,e, what is DDX39A in figure 3. It is explained in the text, but not in the figure caption).

Response: We deeply appreciate the reviewer’s suggestion. According to the reviewer’s comment, we added descriptive legends for Figures 1-5. Thanks again for your professional advice.

Comment 2: Authors could organize and list the papers of Tables 1 and 2 following a general criteria (i.e. “species employed” in Table 1 or “animal model” in Table 2), instead of listing a number of papers.

Response: We appreciate it very much for this insightful suggestion, and we have done it according to your ideas. We reorganized Table 1 by “Species” and Table 2 by “Function”.Thanks again for your professional advice.

Comment 3: In the paragraph of page 5 lines 131-145, authors list a number of studies in which they mention that “a number” of circRNAs where upregulated or downregulated, but they do not analyze if they are a specific circRNA that it is always upregulated or downregulated in all the models, or if a specific family of orthologous genes circRNA plays the same role.

Response: We appreciate it very much for this interesting and meaningful question, and we have added corresponding explanation in the manuscript after extensive literature research and careful consideration.

Comment 4: Page 3 lines 69 and 70. Please, consider to include the description of the different types of PF in the introduction dedicated to “Pulmonary fibrosis” (First paragraph in page 1).

Response: We appreciate it very much for this insightful suggestion, and we have done it according to your ideas.

Comment 5: Page 4 lines 105. When speaking about “making them exceptionally stable once formed “, general readers would appreciate to include the half life of those molecules.

Response: We deeply appreciate the reviewer’s suggestion. According to the reviewer’s comment, we have added the half-life of circRNA and their linear counterparts in the corresponding sections. Thanks again for your professional advice.

Comment 6: Page 8 line 169. Authors should consider to employ the term “prevented” instead of  “reversed”.

Response: We appreciate it very much for this insightful suggestion, and we have done it according to your ideas.

Comment 7: Page 9 line 235. Authors employ , “circular RNA HECT domain E3 ubiquitin protein ligase 1 (circHECTD1))”. They should employ the same criteria when they mention the rest of circRNAs.

Response: We appreciate it very much for this insightful suggestion, and we have done it according to your ideas.

Comment 8: Page 8 line 198. Authors should consider to employ the term “prevented” instead of  “reversed”.

Response: We appreciate it very much for this insightful suggestion, and we have done it according to your ideas.

Comment 9: Page 10 line 264. “Many individual circRNAs have been shown to function by activating fibroblast”. Please, provide a reference/s for this statement.

Response: We appreciate it very much for this insightful suggestion, and we have done it according to your ideas.

Comment 10: Page 12 line 370. “qRT-PCR demonstrated that circHIPK3 is upregulated in the lung tissues of normal people and IPF patients compared with normal people”. Please, check the groups involved in the comparison.

Response: We are extremely grateful to Reviewer for reviewing the paper so carefully and pointing out this problem. We have rethought the argument in depth and adjusted the text in the manuscript.

Comment 11: Page 11 line 319. A few circRNAs have been proposed to influence the process of FMT, including circHIPK3 and circ0044226. Please, provide a reference/s for this statement.

Response: We appreciate it very much for this insightful suggestion, and we have done it according to your ideas.  

Comment 12: There is a number of mistakes in the text:

Page 2 lines 55 to 57 : “CircRNAs are reported to be implicated in the regulation of many different cellular and pathophysiological processes through diverse mechanisms of action”. This phrase is too vague.

Response: We are extremely grateful to Reviewer for reviewing the paper so carefully and pointing out this problem. We have rethought the argument in depth and adjusted the text in the manuscript.

Comment 13: Page 4 lines 118. When authors mention that “some nuclear circRNAs can also regulate gene expression by enhancing the of RNA polymerase II”. They are referring to an increase in the transcription of the gene of RNA polymerase, the protein levels of RNA polymerase, or the activity of the protein?. The phrase is incomplete or confusing.

Response: We are extremely grateful to Reviewer for reviewing the paper so carefully and pointing out this problem. We have rethought the argument in depth and adjusted the text in the manuscript.

Comment 14: Page 4 line 109. There are two “most” in the same phrase. Authors should consider to employ other term/synonym.

Response: We are extremely grateful to Reviewer for reviewing the paper so carefully and pointing out this problem. We have rethought the argument in depth and adjusted the text in the manuscript.

Comment 15: Page 4 line 112. “Secondly, Some circRNAs”. Some, should be “some” (lowercase letters)

Response: We appreciate it very much for this insightful suggestion, and we have done it according to your ideas.  

Comment 16: Page 5 line 133.” Similarly, A total of 10 circRNAs” . A, should be “a” (lowercase letters)

Response: We appreciate it very much for this insightful suggestion, and we have done it according to your ideas.  

Comment 17: Page 8 line 161. The abbreviation “EndMT” is previously described in page 3 line 72.

Response: Thank you for pointing this out, and we have deleted the redundant abbreviations according to your ideas.

Comment 18: Page 8 line 179. The term “epthelial” should be “epithelial”, and the abbreviation is previously described in page 1 line 32.

Response: We apologize for the mistake and we have corrected it according to your ideas.

Comment 19: Page 9 line 196. The term TGF-β should be described for the first time.

Response: We appreciate it very much for this insightful suggestion, and we have done it according to your ideas.  

Comment 20: Page 8 line 203. The term “upregualted” should be “upregulated”.

Response: We apologize for the mistake and we have done it according to your ideas.

Comment 21: Page 9 line 205. Authors employ , “circular ZC3H4 RNA (circZC3H4)”. They should employ the same criteria when they mention the rest of circRNAs.

Response: We appreciate it very much for this insightful suggestion, and we have done it according to your ideas.

Comment 22: Page 9 lines 220-222. . Taken together, these studies demonstrated that the interaction between circRNA and miRNA may exert important functions and provide potential therapeutic targets in pulmonary”. The phrase is incomplete or confusing.

Response: We are extremely grateful to Reviewer for reviewing the paper so carefully and pointing out this problem. We have rethought the argument in depth and adjusted the text in the manuscript.

Comment 23: Page 9 line 245. “Further Gain- and”. Gain, should be “gain” (lowercase letters)

Response: We apologize for the mistake and we have done it according to your ideas.

Comment 24: Page 10 line 261. “macrophage activation”….the abbreviation is previously described.

Response: Thank you for pointing this out, and we have corrected it in the manuscript according to your ideas.

Comment 25: Page 10 line 286. “silica., while...” Please, check the punctuation mark.

Response: We apologize for the mistake and we have deleted the redundant punctuation mark according to your ideas.

Comment 26: Page 10 line 295. circHIPK2 should be “CircHIPK2” (uppercase letters).

Response: We apologize for the mistake and we have done it according to your ideas.

Comment 27: Page 11 line 298. “promoting the expression of , which is” . The phrase is incomplete.

Response: We are extremely grateful to Reviewer for reviewing the paper so carefully and pointing out this problem. We have rethought the argument in depth and adjusted the text in the manuscript.

Comment 28: Page 11 line 337.HBE cells. The source of the cell type should be described.

Response: Thank you for pointing this out, and we have described the source of the cell type in the manuscript according to your ideas.

Comment 29: Page 11 line 347. “Several lines of evidence indicated that injury to, and dysfunction”. The phrase is incomplete or confusing.

Response: We are extremely grateful to Reviewer for reviewing the paper so carefully and pointing out this problem. We have rethought the argument in depth and adjusted the text in the manuscript.

Comment 30: Page 12 line 350. circular RNA 406961 (circ_406961) has been shown to be decreased in human bronchial epithelial cells (BEAS-2B) were exposed to PM2.5 and inhibits PM2.5-induced inflammatory reaction. The phrase is incomplete or confusing.

Response: We are extremely grateful to Reviewer for reviewing the paper so carefully and pointing out this problem. We have rethought the argument in depth and adjusted the text in the manuscript.

Comment 31: Page 12 line 396. Nternal should be “internal”.

Response: We apologize for the mistake and we have done it according to your ideas.

Comment 32: Page 13 line 401. Huamn should be “human”.

Response: We apologize for the mistake and we have done it according to your ideas.

Comment 33: Page 13 line 404. Studies should be “studies” (lowercase letters).

Response: We apologize for the mistake and we have done it according to your ideas.

We have tried our best to improve the manuscript and made some changes in the manuscript. These changes will not influence the content and framework of the paper.

We appreciate for Editors/Reviewers’ warm work earnestly, and hope that the correction will meet with approval. Once again, thank you very much for your comments and suggestions.
